# The Effect of Age on the Evolution of the Stem Profile and Heartwood Proportion of Teak Clonal Trees in the Brazilian Amazon

Mario Lima dos Santos [1,*], Eder Pereira Miguel [1], Leonardo Job Biali [1], Hallefy Junio de Souza [1], Cassio Rafael Costa dos Santos [2] and Eraldo Aparecido Trondoli Matricardi [1]

[1] Department of Forest Sciences, University of Brasilia, Campus Universitário Darcy Ribeiro W/N, Brasilia 70910-900, DF, Brazil; miguelederpereira@gmail.com (E.P.M.); ljbiali@gmail.com (L.J.B.); hallefyj.souza@gmail.com (H.J.d.S.); ematricardi@unb.br (E.A.T.M.)

[2] Department of Forestry Engineering, Federal Rural University of Amazonia, Capitão Poço Campus, Travessa Pau Amarelo W/N, Vila Nova, Capitão Poço 68650-000, PA, Brazil; cassio.santos@ufra.edu.br

* Correspondence: mariolimaeng@gmail.com; Tel.: +55-91-98170-8268

**Abstract:** Stem profile modeling is crucial in the forestry sector, particularly for commercially valuable species like teak (*Tectona grandis* Linn F.), whose value depends on its stem dimensions, heartwood proportion, and age. We proposed a nonlinear mixed-effect model to describe the evolution of the stem and heartwood profiles of clonal teak trees with ages between 4 and 12 years in the Brazilian Amazon. Tapering models were used to estimate the bark, bark-free, and heartwood diameters. Dummy variables were included in each tapering model to estimate each type of diameter and enable compatibility. We used mixed models with age as a random effect in order to improve the accuracy. The Demaerschalk model provided the most accurate and compatible estimates for all three types of stem diameter. Also, age as a random effect significantly improved the model's accuracy by 7.2%. We observed a progressive increase in the heartwood proportion (14% to 34%) with advancing age, while the proportions of bark (23% to 20%) and sapwood (63% to 45%) showed inverse behavior. The growth rate of the heartwood differed from that of the bark volume, emphasizing the importance of considering the age of heartwood maximization when determining the cutting cycle of the species.

**Keywords:** trunk assessment; conicity; tapering; heartwood proportion; dummy variables; mixed models



## 1. Introduction

Teak (*Tectona grandis* Linn F.) is one of the most cultivated forest species worldwide, with approximately 6.9 million ha planted, of which about 99 thousand hectares are planted in Brazil [1–3]. Its wood is highly regarded for its quality, durability, mechanical resistance, and resistance to pathogens, making it one of the most valuable tree species in the global timber market. Consequently, appropriate silvicultural practices in teak plantations are extremely important. Those practices should favor individual productivity over the total wood productivity of the stand, promote the tree's straight shape, minimize bifurcation and knots, and, most importantly, maximize the proportion of heartwood in comparison to sapwood [4,5].

These desirable characteristics of teak trees can be considered quality attributes of stands from such species, and they are important targets for monitoring through continuous forest inventories. Therefore, it is essential to quantify the bark, sapwood, and heartwood volumes in teak trees, aiming to size the final product to be generated [3,6]. Higher proportions of sapwood in the trunks are associated with wood treatability and lower natural resistance, while higher proportions of heartwood result in darker tones, which confer greater aesthetic beauty on the wood and greater natural resistance. On the other hand, larger bark dimensions indicate lower yields in the use of the trunk [3].

In this context, the usage of analytical tools is fundamental to ensure that the standing trees present the heartwood in their stem [7]. Among the various analytical tools, the tapering function is one of the most widespread and promising [6,8,9]. Tapering equations are used to describe the variation of the tree diameter along the trunk, i.e., in terms of height. These equations are useful to estimate the amount of wood in a tree and, consequently, to estimate the volume of wood in a forest stand [10,11]. Their importance lies in the possibility of estimating the diameter at a given height, as well as the volume of sections and portions along the profile. This analysis allows us to know the various uses of the stem that can be assigned to its different parts, allowing for greater yield and use of production [12–14].

In addition to the advantages of flexibility and range of information generated, tapering functions allow for the principle of compatibility between the estimated total and commercial volumes, i.e., the sum of the integral of the partitioned stem volumes must be equal to the total volume [15]. This procedure increases the range of forest products that can be marketed, according to demand. With different products, there is an increase in the production yield and, consequently, higher added value in the price of the wood [6,14].

Tapering models can be linear or nonlinear. Linear models, for being simpler, may often not be able to capture the complexity of the diameter variation along the trunk. On the other hand, nonlinear models can be more flexible but also more complex [16]. Tapering models, in their simple form (i.e., without modifications), are generally less accurate and less realistic in describing tree stem profile compared to nonlinear models. For this reason, several attributes can be used to compose forest stand tapering models, with an emphasis on age, which is one of the variables that most interferes with the shape of tree stems [17,18] and which has not been much explored in this kind of modeling.

This type of assortment analysis is widely used by companies in the planted forest sector for different species worldwide [15,19]. However, for teak clonal plantations, which have great economic importance and are increasing in the forest sector, there is still the need for studies involving tapering functions in addition to those already performed for seminal stands of this species [6,20].

The commercialization of teak wood has diversified its production for laminate, sawn timber, and chips for energy, among other purposes. Thus, in addition to knowing the tapering and assortment of the stem of this species, it is necessary to quantify the dimensions of bark, sapwood, and heartwood, as these are very evident structures and distinguishable by color and formation [21,22]. Among the internal structural variables of the stem, the heartwood is the attribute that adds the greatest value, as it is the portion that is valued by the teak wood market, which is justifiable because of its high resistance, easy workability, and different uses, such as furniture, civil, and naval construction [23–25].

In the current literature, several studies have described the stem profiles of trees in detail, covering various forest species across different regions. However, there is a scarcity of research focused on predicting the dimensions and proportions of heartwood in the stems of these species over time. In this analysis, we formulated the following scientific questions and corresponding hypotheses. Q1: Can the usage of nonlinear models with age as a random effect for modeling teak stem tapering result in more accurate estimates compared to traditional tapering models? H1: Nonlinear tapering models that incorporate age will provide more accurate and meaningful estimates of stem diameters compared to traditional models. Q2: Does the proportion of heartwood in teak clonal stands follow a pattern of evolution in relation to stem growth with and without bark? H2: There is a positive variation in the proportion of heartwood in the total volume as the trees mature. With these research questions and hypotheses, our objective was to propose a nonlinear model with age as a random effect that describes the evolution of the stem and heartwood profiles of clonal teak trees in the eastern Brazilian Amazon. This model aims to estimate diameters over time and quantify the ratio between heartwood and sapwood proportions.

## 2. Materials and Methods

### 2.1. Study Area and Silvicultural Practices

We performed the present study in teak clonal stands implanted in the municipality of Capitão Poço, State of Pará, eastern Amazon, Brazil (from 2°30′00″ S; 47°20′00″ W, to 2°20′0″ S; 47°30′0″ W) (Figure 1). The relief from the region ranges from slightly undulating to flat. The predominant soils are classified as latosol, typical dystrophic yellow latosol, petroplintic dystrophic yellow, and petric concretionary plintosoil [26]. The dominant natural vegetation in this region is dense ombrophylous forest [27]. The climate type can be classified as Am, according to the Köppen classification, and it is characterized as hot and humid, with a short dry season [28]. The region shows an average annual rainfall of 2256 mm and an average temperature of 26.1 °C [29].

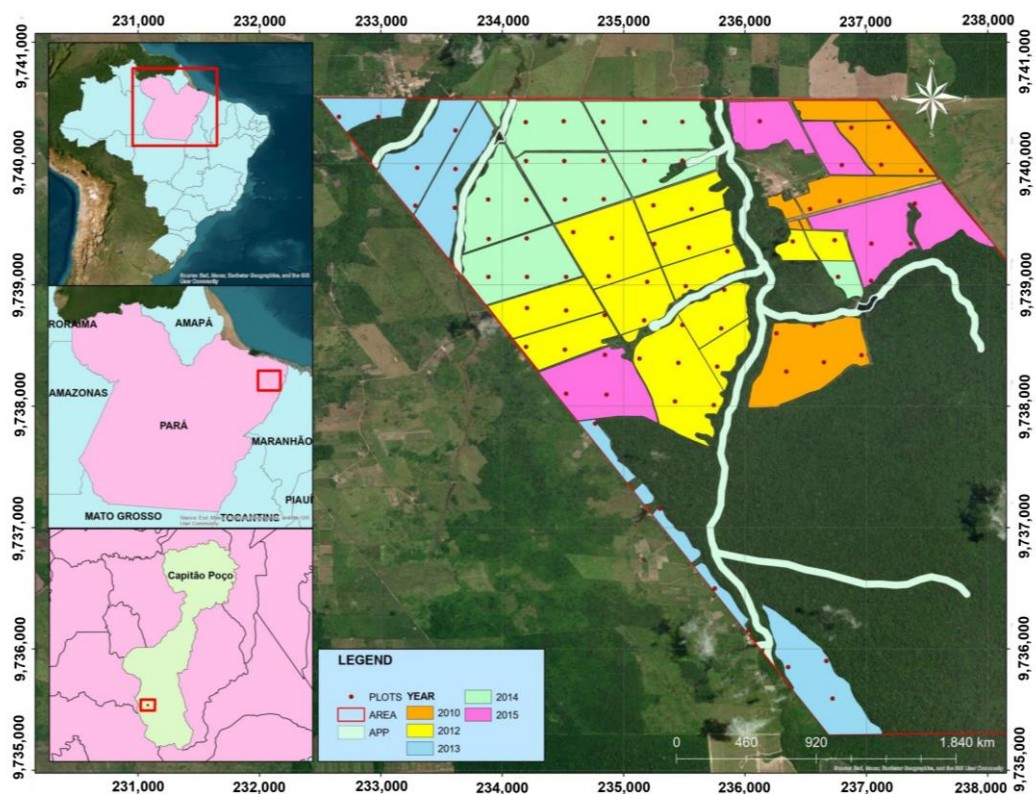

**Figure 1.** Study site location in the Capitão Poço municipality, Pará State, eastern Brazilian Amazon.

The teak clonal plantations were established in the years 2010, 2012, 2013, 2014, and 2015. The seedlings were manually planted at the following spacings: 3.5 × 3.5 m, 3.75 × 3.75 m, and 4 × 4 m. For all stands, the following silvicultural techniques were performed using identical calendars: cleaning the area with a bulldozer, combating leaf-cutting ants with ant bait; liming with dolomitic limestone, applying 3 t ha$^{-1}$; base fertilization, with application of 200 g plant$^{-1}$ of NPK 8-28-16 and 100 g plant$^{-1}$ of KCl, in the planting hole; weed control using crowning with the aid of a hoe, as well as mechanized and semi-mechanized weeding with a hydraulic tractor; coverage fertilization with application of 7 g plant$^{-1}$ of boron and 100 g plant$^{-1}$ of KCl; and artificial pruning with a saw and motor-pruner [30]. Systematic thinning was also performed at 4.5 (1st) and 8.5 (2nd) years, aiming to reduce the basal area by 50% for both thinning practices [30,31].

### 2.2. Stem Assessment

We performed a stem analysis of 121 sample trees, which were selected based on the different diameter classes (number of individuals selected pro-rata to each class) and age (4 to 12 years) of the stands (Table 1). Wood discs approximately 5 cm thick were

taken along the tree stem and measured at orthogonal positions using a 50 cm ruler at 12 positions following the Hohenadl method. We recorded the diameters ($d_{0,i}$) at positions corresponding to 0, 5, 15, 25, 35, 45, 55, 65, 75, 85, and 95% of the tree total height [32] and at 1.3 m from the ground (*dbh*). With this, we obtained the diameters for with bark ($d_{wb}$), without bark ($d_{wob}$), and heartwood ($d_h$).

**Table 1.** Descriptive statistics of trees used for modeling the stem tapering of teak clonal plantations in eastern Brazilian Amazon.

| Variables | Min. | Max. | Mean | σ |
|:---:|:---:|:---:|:---:|:---:|
| $t$ | 4 | 12 | 6.8 | 2.32 |
| $dbh_{wb}$ | 9.65 | 38.95 | 24.36 | 5.88 |
| $dbh_{wob}$ | 7.75 | 35.00 | 21.43 | 5.26 |
| $dbh_h$ | 2.50 | 25.70 | 13.52 | 4.76 |
| $th$ | 8.70 | 20.05 | 16.51 | 2.05 |

$t$: Age (years); $th$: total height (m); $dbh_{wb}$: diameter at 1.3 m from the ground, with bark; $dbh_{wob}$: diameter at 1.3 m from the ground, without bark; $dbh_h$: heartwood diameter at 1.3 m from the ground; σ: standard deviation.

## 2.3. Modeling and Selection of Tapering Models

We applied four nonlinear tapering models for each diameter type ($d_{wb}$, $d_{wob}$, and $d_h$) (Table 2) to the data from 121 trees. These models were constructed considering the diameter at 1.3 m from the ground with bark ($dbh_{wb}$), total height ($th$), and specified height at any point of the stem ($hi$). To ensure consistency among the diameter estimates, we included dummy variables ($Tx_1$ and $Tx_2$), in which $Tx_1 = 0$ and $Tx_2 = 0$ for diameters with bark, $Tx_1 = 1$ and $Tx_2 = 0$ for diameters without bark, and $Tx_1 = 1$ and $Tx_2 = 1$ for heartwood diameters [6]. In our analysis, we defined as commercial those stems whose minimum diameters were 10 cm ($d_{wb}$) and 5 cm ($d_{wb}$). The reason for this definition is that diameters equal to or greater than 10 cm are considered viable for trade. The smaller ones represent the structure of the tree crown, which is not commercialized in the roundwood market, and the splitting of these stems is not viable in the sawmill.

**Table 2.** Tapering models selected to fit the data from clonal teak plantations in the eastern Brazilian Amazon.

| Model No. | Model | Author |
|:---:|:---:|:---:|
| 1 | $d_i = dbh_{wb}\left[\beta_0 + \beta_1\left(\frac{h_i}{t_h}\right) + \beta_2\left(\frac{h_i}{t_h}\right)^2 + \beta_3\left(\frac{h_i}{t_h}\right)^3 + \beta_4\left(\frac{h_i}{t_h}\right)^4 + \beta_5\left(\frac{h_i}{t_h}\right)^5 \exp\left(\left(\beta_6\, Tx_1\,\frac{1}{dbh_{wb}}\right) - \beta_7\, Tx_2\,\frac{1}{dbh_{wb}}\right)\right] + \varepsilon i$ | Schöepfer (1966) |
| 2 | $d_i = \alpha_0 \left(dbh_{wb}{}^{\alpha_1}\right)\left(t_h{}^{\alpha_2}\right)(X_2)^{((\beta_1 X_2{}^4)+(\beta_2(1/\exp(dbh_{wb}/t_h)))+\beta_3(X_2{}^{0.1})+\beta_4(1/dbh_{wb})+\beta_5(t_h{}^{Q}+\beta_6 * X_2))} * \exp((-\beta_7(Tx_1\frac{1}{dbh_{wb}}))-\beta_8(Tx_2\frac{1}{dbh_{wb}}))+\varepsilon i$ <br> $X_2 = \left[1 - \left(\frac{h_i}{t_h}\right)^{1/3}\right]\bigg/\left[1 - \left(\frac{1.3}{t_h}\right)^{1/3}\right]$ <br> $Q = \left[1 - \left(\frac{h_i}{t_h}\right)^{1/3}\right]$ | Kozak (2004) |
| 3 | $d_i = dbh_{wb}\left[\sqrt{\beta_0\left(\frac{(t_h-h_i)^{\beta_1}}{t_h{}^{\beta_1+1}}\right) + \beta_2\left(\frac{(t_h-h_i)}{t_h}\right)^{\beta_3} + \beta_4\left(\frac{(t_h-h_i)^{\beta_5}}{t_h{}^{\beta_5+1}}\right)}\right]\exp\left[\left(-\beta_6\, Tx_1\,\frac{1}{dbh_{wb}}\right) - \beta_7\, Tx_2\,\frac{1}{dbh_{wb}}\right] + \varepsilon i$ | Demaerschalk (1973) |

$di$: Diameter with bark, without bark, or heartwood (cm), being the correspondent at any height; $h_i$: height specified at any section of the stem; $dbh_{wb}$: diameter at 1.3 m from the ground, with bark; $t_h$: total height (m); $\alpha_i$ and $\beta_i$: regression parameters; $\varepsilon i$: random error. Source: [10,16,33,34].

For each diameter type, we chose the model that showed the highest accuracy, as well as a coherent and biologically realistic stem profile. We fitted the tapering models by cross-validation using the *K-fold* approach, which is recommended in [35], with 10 groups. In [36], grouping the data into 10 parts was the best combination for the model selection. The method consisted of dividing the data into 10 equal parts. For this purpose, the method used 9 parts of these data for the adjustment, with the remainder for validation. Then, the results were combined by obtaining the averages of the following statistical criteria: highest coefficient of Pearson's linear modification between the observed and predicted values ($r_{\hat{y}y}$),

lowest root mean square error (*RMSE*), lowest root mean square error (*RMSPE*) [37], and lowest Akaike's information criterion (*AIC*) [38]. We carried out a graphical analysis of the distribution of the percentage residuals for the observed and estimated values, calculated as the difference between the real and estimated values, divided by the real value multiplied by one hundred, with the significance of the regression parameters and the normality of the residuals using the Shapiro–Wilk test at a 95% probability [39].

We fitted the models using the nonlinear generalized least squares method utilizing the "*gnls*" function from the "*nlme*" package in R® Studio software, 4.3.1. version [40]. To assess any potential autocorrelation, we included a continuous first-order autoregressive error structure in the error term, which enabled the application of the models to longitudinal and unbalanced datasets [41]. The residuals were considered to have an autoregressive structure, and the models were fitted separately [42]. After selecting the most accurate model that efficiently described the tapering for each kind of diameter ($d_{wb}$, $d_{wob}$, and $d_h$), we applied mixed-effect modeling with age (t) as a random effect, thus testing the first hypothesis.

### 2.4. Volume Estimation and Increase

Because of the absence of defined integration in certain models for estimating the stem volume for each diameter type, we resorted to the numerical integration process. To carry out this numerical integration, we reformulated the tapering function as a function of the dependent variable, "$d_i$", where applicable. Subsequently, we squared this function and multiplied it by the constant: "$\pi/40{,}000$". This transformation allowed us to utilize the variable "$d_i$" in centimeters and obtain the volume result in cubic meters [34].

The integration of the function was performed between two specified limits, employing 100 subdivisions within these limits for the integration process. Unlike the conventional method of calculating the height of a cylinder by multiplying the height by the sectional area, we did not apply this step in our numerical integration technique for the tapering function. This omission was unnecessary, as the height is already implicitly accounted for within the tapering function. The integration process was defined as follows:

$$v = \int_{l_i}^{l_s} \frac{\pi}{40000} w^2 \, di \, w$$

where $v$ is the estimated stem volume (m$^3$); $l_i$ is the lower limit used in the integration process; $l_s$ is the upper limit used in the integration process; $d_i$ is the diameter with bark, without bark, or heartwood (cm); and $w$ is the tapering model as a function of the dependent variable, *di* [34].

After calculating the volume through the integral of each tapering model, we proceeded to analyze the proportions and calculate the increments of volumetric production to test the second hypothesis. The mean annual increment (*MAI*) was determined by dividing the current volume (*v*) by the tree age (*t*) (1). In addition, we calculated the current annual increment (*CAI*), defined as the difference between the current volume (*v*) and the volume at the previous age ($v_0$) (2). A schematic representation of the data collection and analysis procedures is shown in Figure 2.

$$IMA = \frac{v}{t} \tag{1}$$

$$CAI = v - v_0 \tag{2}$$



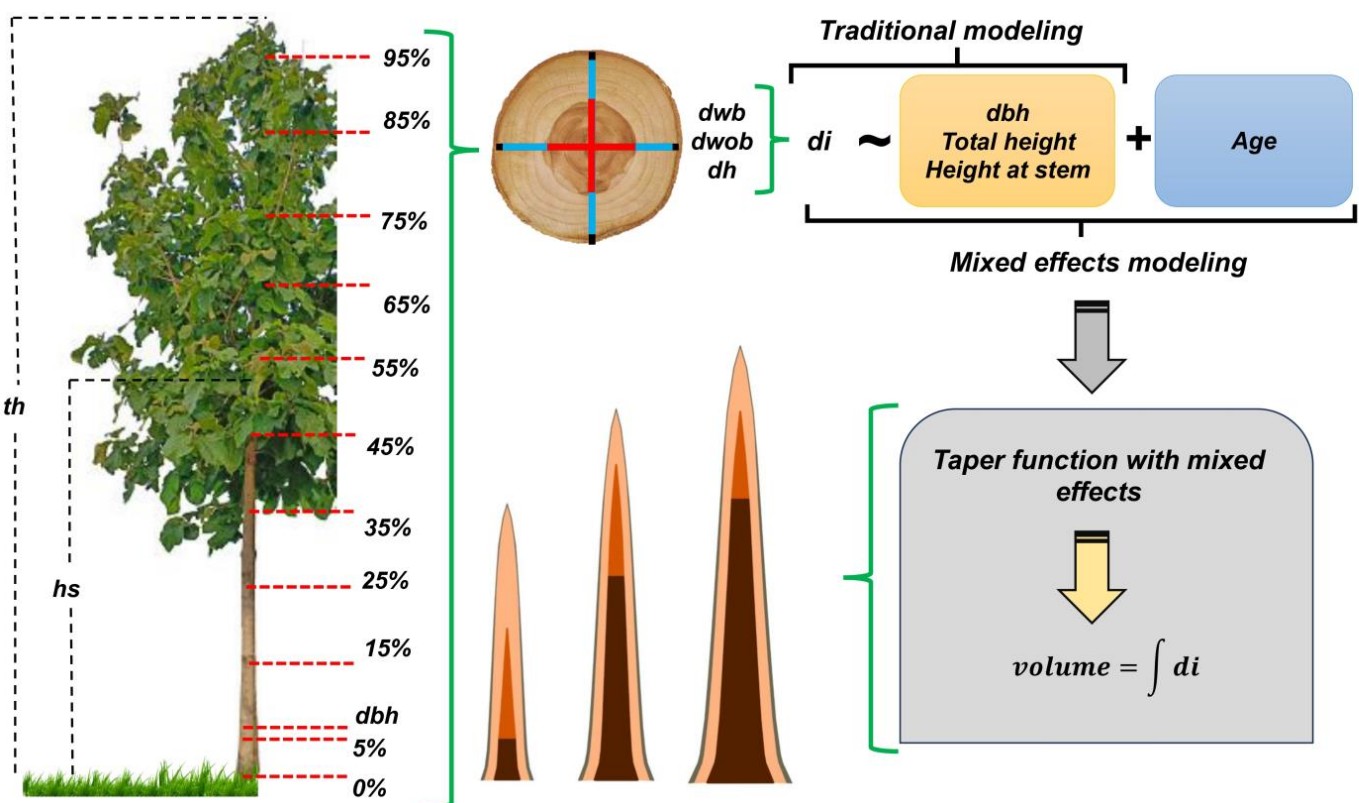

**Figure 2.** Schematic representation of the sequential procedures for collecting and modeling the stem and heartwood profiles of teak clonal plantations in the eastern Brazilian Amazon.

## 3. Results

### 3.1. Tapering of the Stem-Forming Structures

The selection criteria indicated that the Demaerschalk (M3) tapering model was the most accurate of the models fitted for the three types of diameters, with the lowest estimation errors for diameters $RMSE$ = 1.532 cm and $RMSPE$ = 8.634% (Table 3). The accuracy of this model was also confirmed by the lowest $AIC$ value (6872.109). In addition, the correlation coefficient generated using the Demaerschalk model was 0.978, which indicates a strong degree of relationship between the estimated and observed values of the diameters along the trunk structures. Kozak's model was the one with the quality of fit measures closest to those of Demaerschalk's model. Demaerschalk's mixed model generated the following errors for each type of diameter: diameter with bark, with an $RMSE$ = 1.305 cm and $RMSPE$ = 5.945%; diameter without bark, with an $RMSE$ = 1.415 cm and $RMSPE$ = 7.321%; and heartwood diameter, with an $RMSE$ = 1.534 cm and $RMSPE$ = 12.821%.

Based on the results obtained, we selected the Demaerschalk model for the mixed-effect modeling, with age as the random effect, in order to obtain more accurate estimates of stem diameters. The random effect was inserted into parameters $b_0$, $b_2$, $b_4$, and $b_7$, generating a better combination that indicated the significance of the parameters and greater precision, with realistic estimates of the types of diameters at different ages (Table 4). After adjusting the model, there was an average gain in accuracy of 7.2%, with 7.9%, 7.4%, and 5.4% for the diameters with bark, without bark, and heartwood, respectively, with the bark diameter being more influenced by age as a random effect.

**Table 3.** Estimators and precision statistics of the thinning models for estimating each type of diameter, with the models modified with dummy variables and with a random age effect, for clonal teak plantations in the Brazilian eastern Amazon.

| Model No. | Model | Parameters | Standard Error of the Parameters | $r_{\hat{y}y}$ | RMSE (cm) | RMSPE (%) | AIC |
|---|---|---|---|---|---|---|---|
| M1 | Schöepfer (1966) | $b_0 = 1.33336$ ** <br> $b_1 = -7.03844$ ** <br> $b_2 = 42.12448$ ** <br> $b_3 = -131.16996$ ** <br> $b_4 = 189.42682$ ** <br> $b_5 = -102.60384$ ** <br> $b_6 = 3.44771$ ** <br> $b_7 = 10.68827$ ** | 0.00562 <br> 0.153444 <br> 1.678178 <br> 7.104489 <br> 12.618292 <br> 7.912444 <br> 0.124378 <br> 0.181427 | 0.975 | 1.680 | 9.466 | 7015.47 |
| M2 | Kozak (2004) | $\alpha_0 = 0.91005$ ** <br> $\alpha_1 = 0.93388$ ** <br> $\alpha_2 = 0.11368$ ** <br> $b_1 = 46.68804$ ** <br> $b_2 = -11.19448$ ** <br> $b_3 = 98.48906$ ** <br> $b_4 = 40.37705$ ** <br> $b_5 = 0.40839$ ** <br> $b_6 = -142.92871$ ** <br> $b_7 = 3.26184$ ** <br> $b_8 = 10.91128$ ** | 0.05794 <br> 0.01907 <br> 0.03731 <br> 2.1297 <br> 3.89455 <br> 4.26556 <br> 32.4903 <br> 0.11383 <br> 6.23398 <br> 0.11161 <br> 0.16274 | 0.977 | 1.558 | 8.783 | 7036.914 |
| M3 | Demaerschalk (1973) | $b_0 = 12.687462$ ** <br> $b_1 = 1.193167$ ** <br> $b_2 = 0.753512$ ** <br> $b_3 = 28.045142$ ** <br> $b_4 = 0.017068$ ** <br> $b_5 = 3.822944$ ** <br> $b_6 = 3.331303$ ** <br> $b_7 = 10.744412$ ** | 0.5258129 <br> 0.0496739 <br> 0.0351115 <br> 2.0379679 <br> 0.0023329 <br> 0.6839145 <br> 0.1112617 <br> 0.162351 | 0.978 | 1.532 | 8.634 | 6872.109 |
| M4 | Demaerschalk (1973) Mixed-age model | $b_0 = 12.397079$ ** <br> $b_1 = 1.146381$ ** <br> $b_2 = 0.72975$ ** <br> $b_3 = 30.008968$ ** <br> $b_4 = 0.019715$ ** <br> $b_5 = 3.905267$ ** <br> $b_6 = 3.343096$ ** <br> $b_7 = 10.693234$ ** | 0.5906596 <br> 0.0468075 <br> 0.0554166 <br> 1.8642629 <br> 0.0021929 <br> 0.5021687 <br> 0.1038175 <br> 0.4430965 | 0.981 | 1.429 | 8.055 | 7381.798 |

$b_i$: Model parameters; *RMSE*: root mean square error (cm); *RMSPE*: root mean square percentage error; $r_{\hat{y}y}$: correlation coefficient between observed and estimated values; *AIC*: Akaike information criterion. Regression parameters were significant at a 99% (**) probability, using the Student's *t*-test. Mn: Model 1, 2, 3, 4.

**Table 4.** Estimators of the random effect of the Demaerschalk thinning model for estimating diameters with bark, without bark, and heartwood for clonal teak plantations in the Brazilian eastern Amazon.

| Age (Years) | $b_0$ | $b_2$ | $b_4$ | $b_7$ |
|---|---|---|---|---|
| 4 | −0.4100792 | −0.005000337 | 0.00145214 | −2.02136461 |
| 5 | −0.5018386 | 0.041447436 | 0.001243149 | −0.96724703 |
| 6 | 0.0723395 | −0.009890323 | −0.000132658 | 0.01149478 |
| 7 | 0.6856371 | −0.111748992 | −0.00107921 | −0.43425561 |
| 8 | 0.1880284 | −0.014002156 | −0.00048262 | 0.41547726 |
| 9 | 1.1536578 | −0.119782213 | −0.002578956 | 1.45521864 |
| 10 | 0.3420175 | −0.036537151 | −0.000753212 | 0.3976666 |
| 11 | 0.2428273 | −0.026157271 | −0.00053234 | 0.27544133 |
| 12 | −1.7725899 | 0.281671007 | 0.002863706 | 0.86756865 |

$b_i$: Random-effect model parameters.

The Schöepfer model showed the largest errors (*RMSE* and *RMSPE*) and, consequently, biased estimates, with errors of approximately 1.60 cm for the three types of diameters. The order of the accuracy of the models (Demaerschalk > Kozak > Schöepfer) was observed for each type of diameter estimated. This result demonstrates the growth pattern of the variables that make up the stem of teak trees, confirmed by the high correlation among the bark, bark-free, and heartwood diameters along the stem. These results indicate that the Demaerschalk model was the most suitable for accurately estimating the stem diameters of the teak trees studied, with Equation (3), with age as a fixed effect, used for the estimates of each type of diameter.

$$d_i = dbh_{wb}\left[\sqrt{12.40\left(\frac{(t_h-h_i)^{1.15}}{t_h^{1.15+1}}\right) + 0.73\left(\frac{(t_h-h_i)}{t_h}\right)^{30.01} + 0.02\left(\frac{(t_h-h_i)^{3.91}}{t_h^{3.91+1}}\right)}\right]\exp\left[\left(-3.34\,Tx_1\frac{1}{dbh_{wb}}\right) - 10.69\,Tx_2\frac{1}{dbh_{wb}}\right] \quad (3)$$

The distribution of the percentage residuals in the Demaerschalk models with the type of diameter as a dummy variable showed no bias in the estimates, with average residual amplitudes of 0.89 (Figure 3), as did the mixed model 0.21% (Figure 4). These models also showed a high correlation between the estimated and observed values (>0.97) (Figures 3 and 4) and residuals concentrated in the central error classes (Figures 3 and 4). These values represent a decrease in relation to the previous models, indicating a better adjustment of expectations when age is considered as a random effect. These results show the gain in accuracy of including age as a factor in the modeling of diameters, since it contributes to a more appropriate distribution of residues.

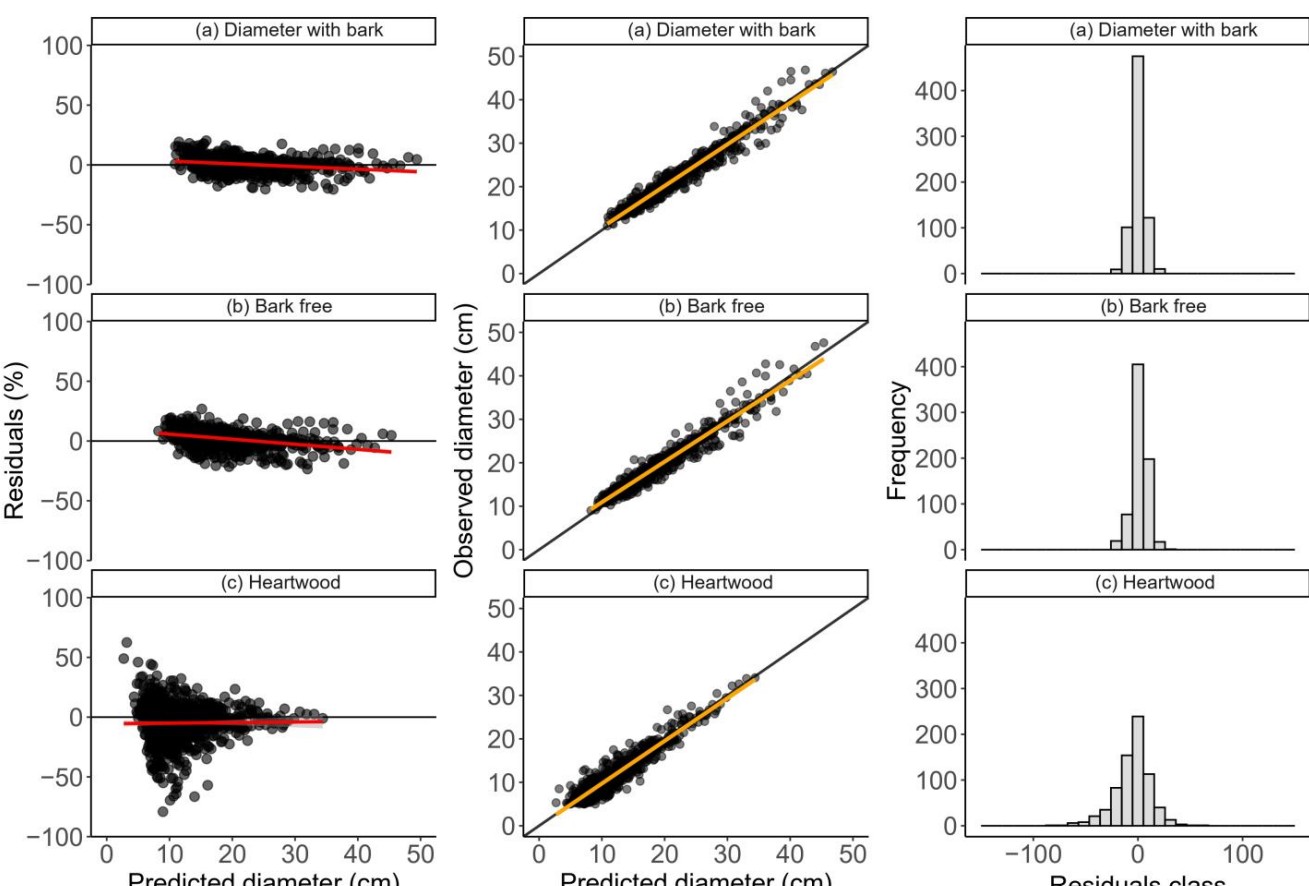

**Figure 3.** Distributions of the percentage estimation errors, correlations between the observed and predicted diameters, and histograms of the relative frequency of errors of the Demaerchalk original tapering model with dummy variable, adjusted to data from clonal teak plantations in the Brazilian eastern Amazon. The red and orange lines (LOESS and linear regression, respectively) represent the data tendency lines.

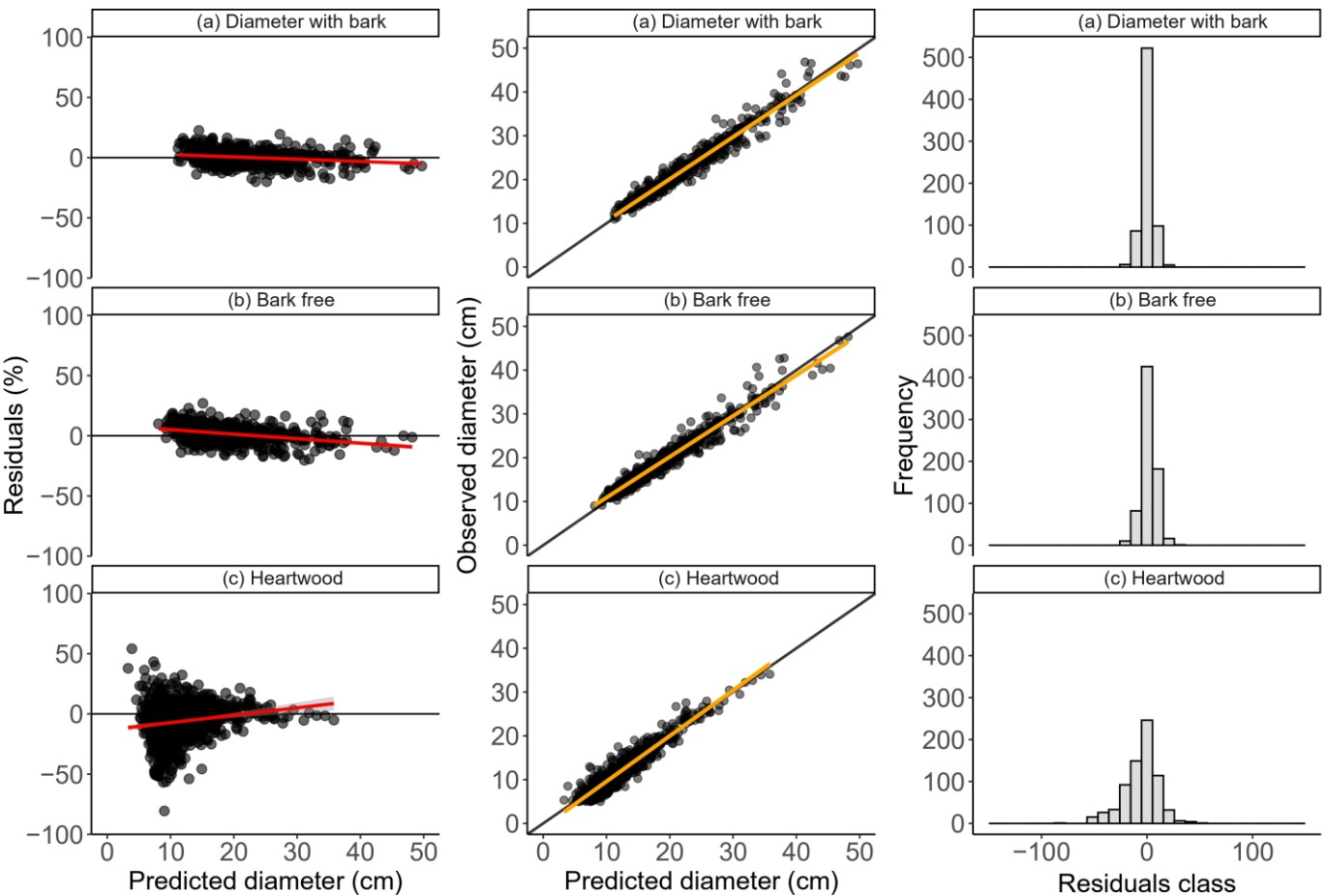

**Figure 4.** Distributions of the percentage estimation errors, correlations between the observed and predicted diameters, and histograms of the relative frequency of errors of the Demaerchalk mixed-tapering model with age as a random effect variable, adjusted to data from clonal teak plantations in the eastern Brazilian Amazon. The red and orange lines (LOESS and linear regression, respectively) represent the data tendency lines.

### 3.2. Diameter Estimation and Volume from Stem-Forming Structures

Using the Demaerschalk mixed model, we generated estimates of each type of diameter forming the trunk structures at different ages. Figure 5 illustrates the diameters with and without bark, as well as the heartwood diameters. Notably, we observed a progressive increase in heartwood diameter as the trees advanced in age, at a much higher rate compared to the growth of bark and sapwood. In addition to the changes in diameter, we also saw an evolution in the position of the heartwood formation in relation to the ground, which means that the height of the heartwood also varied with age.

Figures 6 and 7 describe the stem proportional composition of teak clonal trees over time regarding heartwood, sapwood, and bark for the total and commercial volumes. For the total volume, we observed a progressive increase in the heartwood production. During the period from 4 to 12 years of age, the proportion of the total bark volume decreased from 23% to 20%, and the sapwood volume decreased from 63% to 45%. Heartwood, on the other hand, showed a considerable increase, from 14% to 34% of the total volume. These results show the constant and consistent growth of heartwood over time, with an average rate of increase of approximately 2.8% per year.

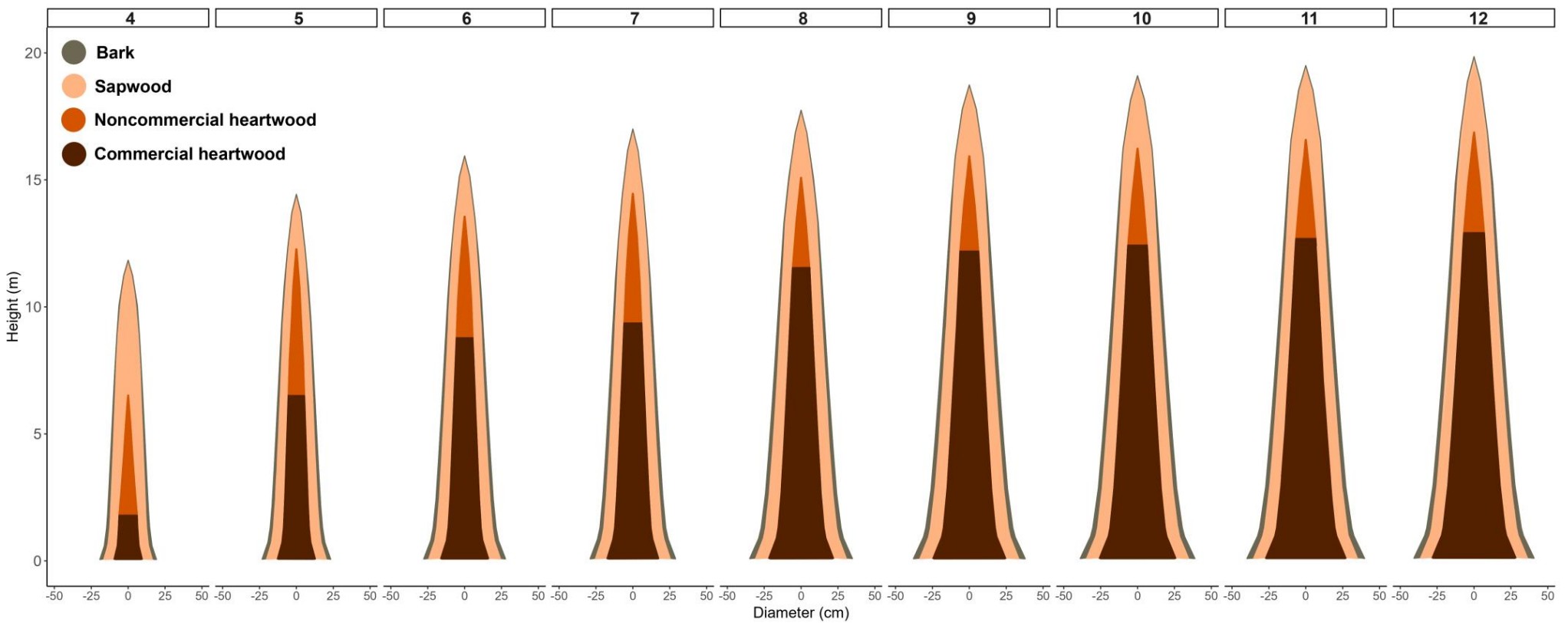

**Figure 5.** Evolution of the trunk-forming structures of teak clonal trees between 4 and 12 years of age estimated using the Demaerschalk mixed model in teak clonal trees in the Brazilian eastern Amazon.

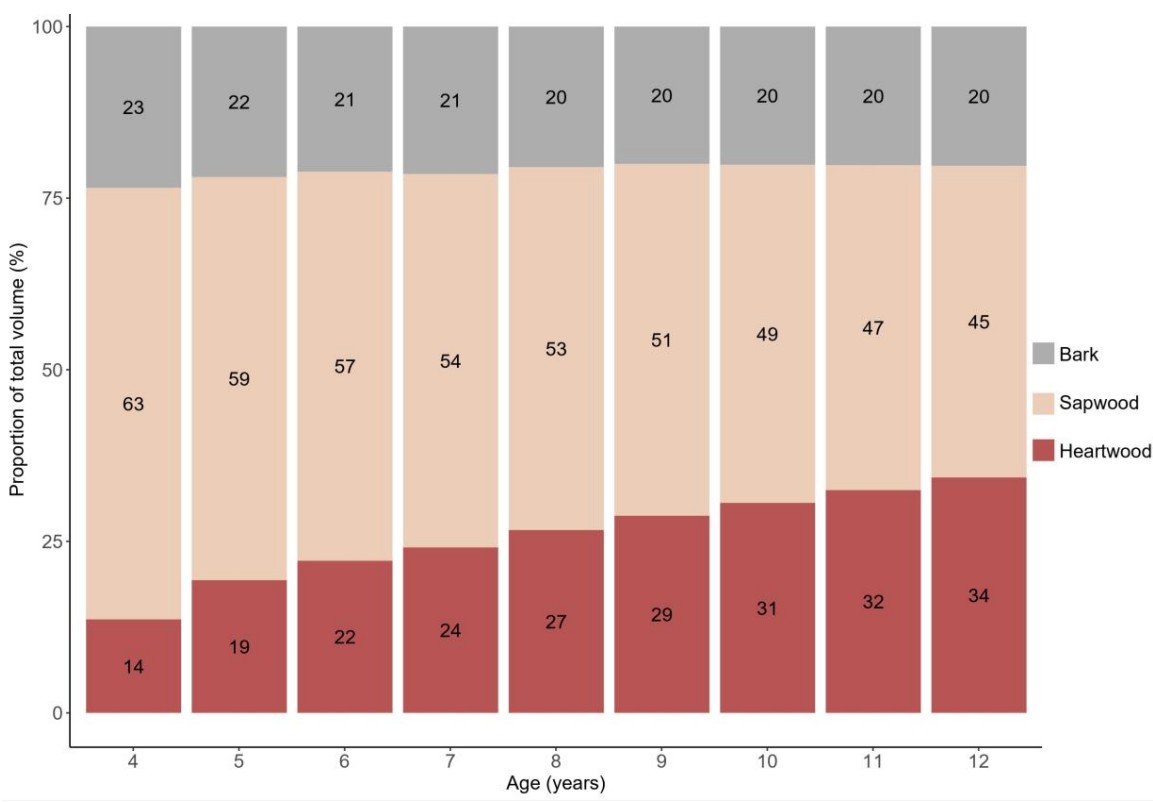

**Figure 6.** Evolution of the proportions of bark, sapwood, and heartwood, with ages between 4 and 12 years, in relation to the total volume with bark, of trees in teak plantations in the eastern Brazilian Amazon.

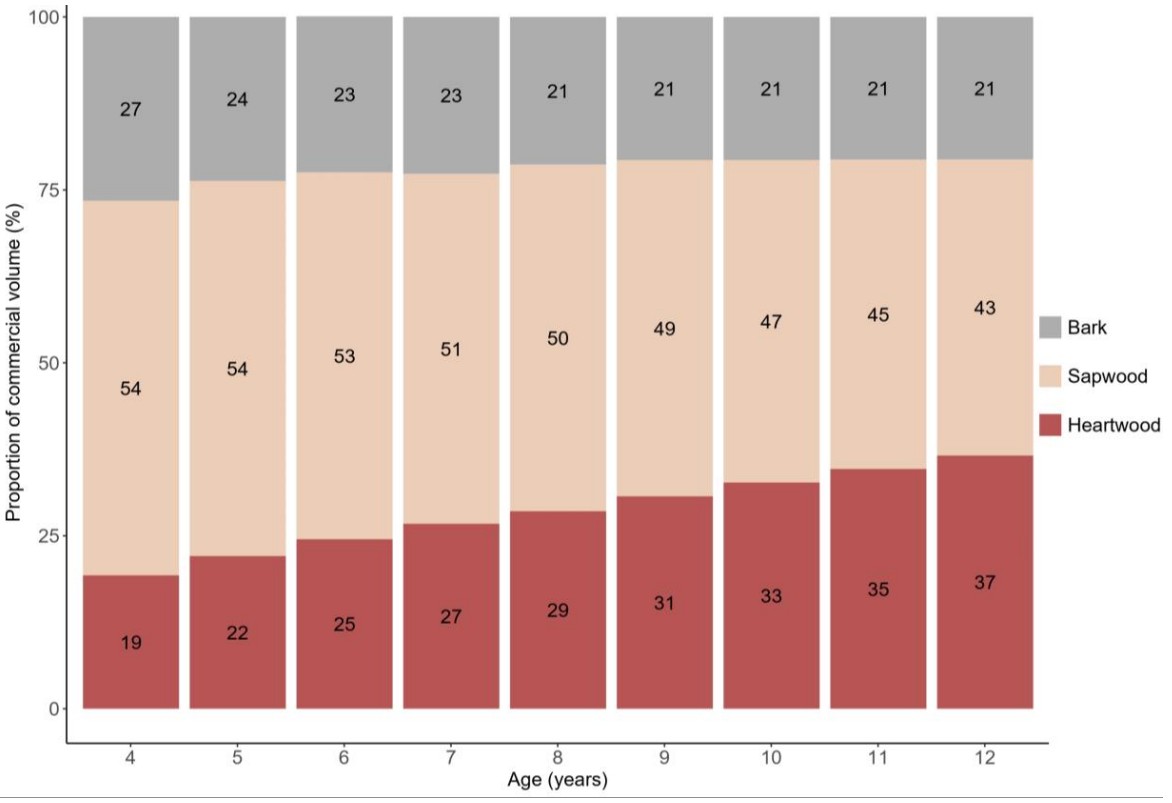

**Figure 7.** Evolution of the proportions of bark, sapwood, and heartwood, with ages between 4 and 12 years, in relation to the commercial volume with bark, of trees in teak plantations in the eastern Brazilian Amazon.

When specifically examining the evolution of commercial wood production in teak stems, we observed a significant variation in the proportions of bark, sapwood, and heartwood volume over time (see Figure 8). During the period from 4 to 12 years of stand age, the proportion of bark volume decreased from 27% to 21%, and sapwood decreased from 54% to 43%. Conversely, the proportion of heartwood increased from 19% to 37% of the total volume. These findings underscore a consistent trend of increasing heartwood content in teak tree stems as they mature.

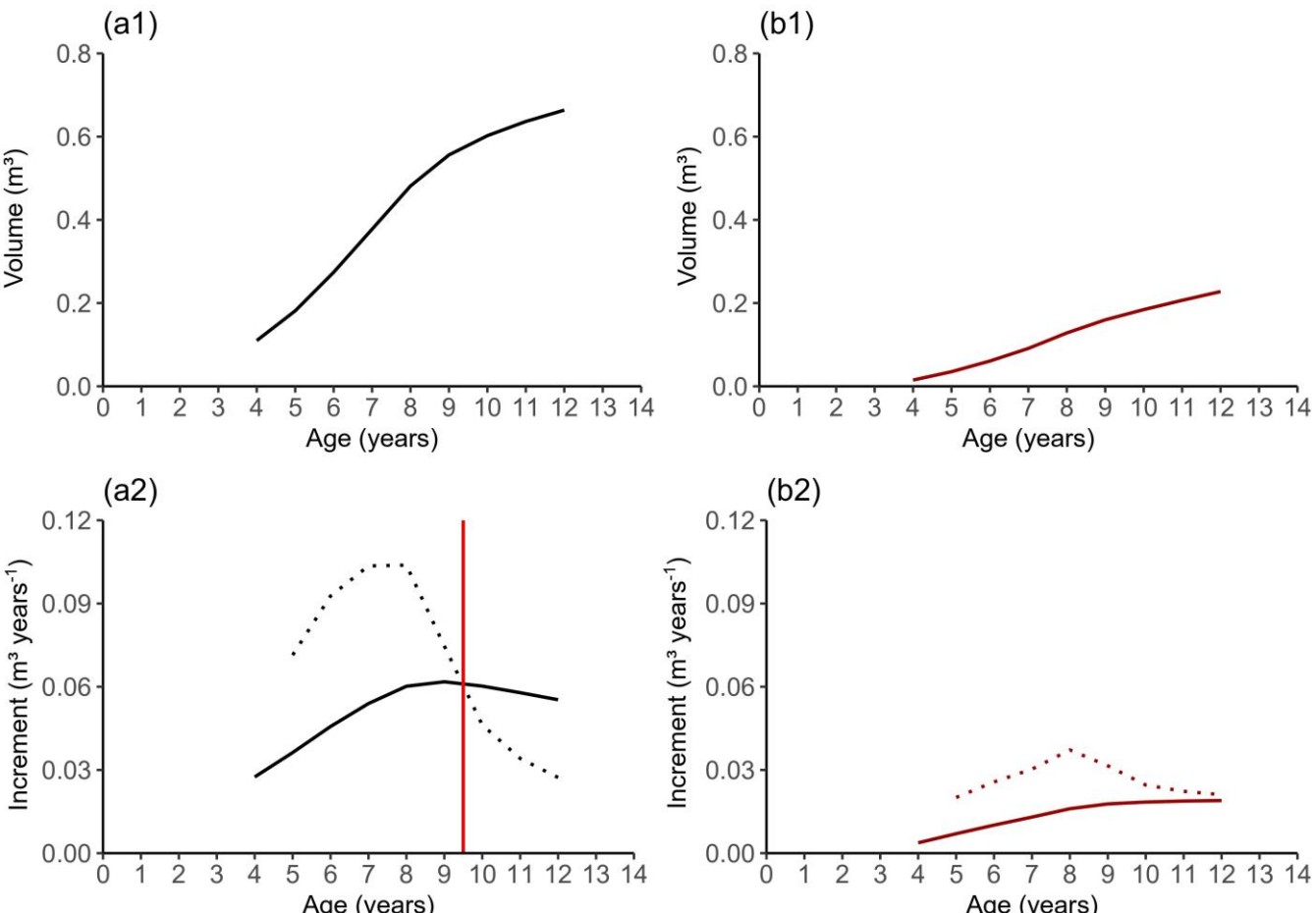

**Figure 8.** Production (1) and increment (2) in bark volume and heartwood volume of teak clonal trees in the eastern Brazilian Amazon. In graphs (**a1**) and (**b1**), the continuous line represents volumetric production of stem with bark and heartwood, respectively. In graphs (**a2**) and (**b2**), the dotted lines represent the CAI, and the continuous lines represent the MAI. The continuous line in red, represents the moment when the MAI and CAI curves cross.

Table 5 presents information on teak stem growth, including measurements for both bark and heartwood. It includes data on the total and maximum heartwood heights concerning age and specific trees, along with the corresponding volumes for bark and heartwood, as well as their respective increments. Over the 12-year period, which is close to the rotation age for clonal teak stands, we observed an average increase of 0.0553 $m^3 \cdot year^{-1}$ in bark volume and 0.0190 $m^3 \cdot year^{-1}$ in heartwood volume. Notably, there was a noticeable difference in the growth rates of the bark and heartwood volumes. Specifically, we observed an inflection point in the growth of the bark volume between ages 9 and 10 years. In this interval, the curves for the bark volume increment (*CAI*) and maximum annual increment (*MAI*) crossed, indicating a reduction in the growth rate (Figure 8). On the other hand, the growth of heartwood did not show any signs of stagnation until the trees reached 12 years

of age. This finding supports the notion of a distinct and independent development of heartwood compared to the rest of the tree stem.

**Table 5.** Increases in the bark and heartwood volumes of average trees by age estimated with tapering modeling for clonal teak plantations in the eastern Brazilian Amazon.

| Age (Years) | With Bark | | | | | Heartwood | | | |
|---|---|---|---|---|---|---|---|---|---|
| | *th* | *dbh* | *Vwb* | *CAI Vwb* | *MAI Vwb* | *hh* | *Vh* | *CAI Vh* | *MAI Vh* |
| 4 | 11.84 | 14.31 | 0.1099 | - | 0.0275 | 6.51 | 0.0150 | - | 0.0037 |
| 5 | 14.43 | 17.78 | 0.1813 | 0.0714 | 0.0363 | 12.27 | 0.0351 | 0.0201 | 0.0070 |
| 6 | 15.94 | 21.46 | 0.2740 | 0.0927 | 0.0457 | 13.55 | 0.0607 | 0.0257 | 0.0101 |
| 7 | 17.00 | 22.51 | 0.3776 | 0.1036 | 0.0539 | 14.45 | 0.0910 | 0.0303 | 0.0130 |
| 8 | 17.74 | 27.22 | 0.4814 | 0.1038 | 0.0602 | 15.08 | 0.1282 | 0.0372 | 0.0160 |
| 9 | 18.74 | 29.61 | 0.5561 | 0.0747 | 0.0618 | 15.93 | 0.1597 | 0.0315 | 0.0177 |
| 10 | 19.10 | 30.48 | 0.6023 | 0.0462 | 0.0602 | 16.24 | 0.1843 | 0.0246 | 0.0184 |
| 11 | 19.50 | 31.36 | 0.6364 | 0.0341 | 0.0579 | 16.58 | 0.2066 | 0.0223 | 0.0188 |
| 12 | 19.85 | 32.23 | 0.6637 | 0.0273 | 0.0553 | 16.87 | 0.2277 | 0.0211 | 0.0190 |

*th*: Tree total height (m); *hh*: heartwood maximum height in the tree (m); $dbh_{wb}$: diameter at 1.3 m from the ground, with bark (cm); *Vwb*: total volume with bark from the average tree ($m^3$); *Vh*: heartwood total volume from the average tree ($m^3$); *CAI*: current annual increase in volume ($m^3$ $year^{-1}$); *MAI*: mean annual increase in volume ($m^3$ $year^{-1}$).

## 4. Discussion

The Demaerschalk model was the most accurate in estimating the structures of the stem profiles in clonal teak trees with bark, without bark, and heartwood, especially when including age as a random effect in the mixed models assessed, with errors of 1.305, 1.415, and 1.534 cm. A study conducted in seminal teak plantations in the State of Mato Grosso, Brazil, covering trees with ages between 3 and 12 years, recommended the Garay model [6]. This model presented errors (1.34 cm with bark, 1.35 cm without bark, and 2.10 cm heartwood) close to those found in our work, except for heartwood diameter. In the study conducted in [8], in the Baixo Rio Acre microregion, Acre State, Brazil, the authors investigated the profile of stems with bark of seminal teak trees, with ages ranging from 6 to 10 years. During the research, the authors used the model in [43], which provided favorable results to describe the stem profile. However, they found that this model had a percentage error of more than 8%, which is 2% more than the error found for our study (6%). It is important to note that both studies cited above did not test any mixed models with age as a random effect.

The study carried out in [17] investigated the impact of age on the parameters of a fifth-degree polynomial (Schöepfer model). This study was used to describe the profile of the stem with bark of trees of the species *Pinus taeda* L. in the midwest region of the state of Santa Catarina, Brazil. The authors examined eight decomposition combinations and found that the modification of the $b_1$ and $b_5$ coefficients resulted in the best statistical performance, biological realism, and consistency of the trunk shape of these trees. The results achieved through our research, which involved an in-depth analysis of teak clonal trees, revealed that the performance of the Demaerschalk model (without the inclusion of random effects) showed lower levels of accuracy. However, once we implemented the mixed-effects modeling approach, we observed notable improvements in the predictive capacity of the Demaerschalk model, particularly with regard to the prediction of tree heartwood characteristics. These findings reinforce the importance of the adequate decomposition of model parameters according to the age of the studied trees, since the modification of specific parameters resulted in more accurate models and in conformity with the real characteristics of the trunks of the species.

The findings of our research suggest that the Demaerschalk model can be a suitable and reliable tool for estimating stem diameters in clonal teak plantations, leading to more accurate results when tree age is considered. Based on this, the results of the present study did not indicate evidence to reject the first hypothesis elaborated, which claims that

mixed models that include age as a random effect in their fitting are more accurate and significant in estimating the diameters of the stem-forming structures of clonal teak trees. This inclusion of age as a random effect in the mixed models provided better statistical indicators compared to conventional models, especially for the estimation of with bark diameter (7.9% gain).

There are several factors that influence the heartwood profile and formation in teak wood. Studies, such as those performed in [4,21], highlight age, longitudinal variation, geographical location, environmental conditions, and silvicultural activities as factors of great importance that affect the extractive content of heartwood, color variation, and wood durability. Furthermore, genotypic and phenotypic factors, linked to more or less homogeneous genetic material, also influence the production and proportion of heartwood to sapwood. These studies highlight the complexity and diversity of elements that contribute to heartwood characteristics in teak trees. In addition, ref. [44] found that the heartwood proportion varies significantly among trees depending on the ecological zone in which they are located. These findings highlight the influence of ecological and structural factors on heartwood formation in teak trees.

Several studies have investigated the influence of age on heartwood growth in teak trees, offering valuable insight into this subject. Ref. [45] found that the heartwood/sapwood ratio, as well as the extractive content, tends to increase with age. In addition, ref. [4] noted that factors such as age, location, and environmental conditions can affect the heartwood extractive content, color variation, and wood durability. This highlights the need for considering multiple factors in studies on heartwood growth in teak trees. Another relevant study was conducted in [22], which found that heartwood formation starts at 4 years of age and progressively increases with time. Finally, ref. [46] observed that both cambial activity and wood structure are influenced by tree age and rainfall. Taken together, these studies suggest that age is a major factor to be considered when studying heartwood growth in teak trees. This corroborates the positive results found in our study for the modeling of tapering, considering the inclusion of age as a covariate, especially in the description of the heartwood profile.

The proportion of heartwood volume found in the stem of clonal teak trees in this study is in line with or below the values found by other studies. In the Brazilian context, the research carried out in [6], when evaluating teak trees at 12 years of age, found that approximately 50% of the commercial volume with bark corresponded to heartwood. In other countries, technical rotation periods are longer when compared to plantations in Brazil. In Costa Rica, this resulted in heartwood proportions of 16% at 12 years and 61% at 47 years [47], which were lower than the 34% reported in our study at 12 years. In Bolivia, ref. [48] observed that an 8-year-old teak plantation had a heartwood proportion of 28% in relation to the bark-free volume. Our results for commercial volumes (29%) and total volumes (27%) with bark were close to those found by these authors.

In addition to age, ref. [47] also observed higher heartwood contents in dry areas (those experiencing 5 months with precipitation less than 100 mm) than in humid zones. This result may justify the higher proportion of heartwood (>50%) in teak seedlings located in the midwest region of Brazil in comparison to our study (37%), as this region has better climate and soil conditions for the development and yield of the species [6], with defined drought seasons. Moreover, it is important to highlight that improved genetic material responds well to growth when compared to seminal trees, presenting high growth rates when total production with bark is evaluated. However, heartwood development and formation resulted in lower heartwood proportions, indicating that its growth rate is lower.

Higher heartwood contents occur at older ages in teak trees, confirming the behavior of heartwood evolution in our study. These variations may be influenced by other factors, such as different spacing, thinning regimes, growth rate, genetic origin, and edaphoclimatic variations [6,44]. In sites with a lower tree density, lower intraspecific competition may favor heartwood formation. Therefore, thinning in these stands tends to favor an increase in heartwood proportions [47]. Dominant and codominant trees are more efficient at

producing heartwood than subdominant and dominated trees, which reinforces the idea of interventions with selective thinning of suppressed trees in order to favor the development of dominant trees. Thus, moderate and heavy thinning tends to produce a higher percentage of heartwood volume [5,47].

Our results reveal a significant increase in the heartwood proportion at younger ages (4, 5, and 6 years), followed by a stabilization in the growth rate from 6 years onwards. These results are corroborated by those obtained in [44], which observed higher growth rates of the heartwood proportion with an increasing trunk diameter in young trees. According to the observations in [44] in West Africa, specifically in Togo, the proportion of heartwood is strongly influenced by the ecological zone in which the trees are located, and, thus, it is an important factor to be analyzed. In cases in which there are sites with significantly different characteristics, the age of the trees alone may not be a reliable determinant to predict whether there will be a higher or lower amount of heartwood. In plantations where no thinning is carried out, the relationships between *dbh* and age, as well as between heartwood volume and age, may not be significant. This finding was confirmed in [44], which was a study of teak plantations at 40 years of age without thinning practices in which a slow heartwood growth was observed, and its formation was not significant with the evolution of age.

Regarding the larger diameters, we observed the evolution of the amount of heartwood of teak trees with age in the teak clonal plantations studied. On the other hand, the amount of sapwood bark decreased, which is an expected biological behavior during the stem formation and development of the species, as stated in [6,49]. In general, the tendency is for bark to decrease as the tree age increases [47]. A higher proportion of heartwood in trees can be advantageous for sawn timber production, especially when the appearance of the wood is an important requirement to be considered, which is an indispensable characteristic of teak wood [3]. In addition, because it is the physiologically inactive part of the wood, the heartwood has higher lignin contents, which gives it greater mechanical strength and resistance to pitting, characteristics that are also essential to produce sawn timber for furniture and shipbuilding [4]. The value of a piece of teak wood is influenced by the proportion of heartwood contained in that piece, stratifying the quality and value of the wood for commercialization [3,6].

It is possible that the favorable environmental conditions and the genetic material (clonal material, which tends to be more productive) in our study provided an accelerated growth rate for teak trees in terms of biomass production, which had positive reflections on the productivity of wood with bark. This fact was confirmed in [31], in which it was suggested that the final harvest in clonal teak plantations in the Brazilian Amazon should be carried out between 13.9 and 16.6 years at the sites with the highest and lowest productive capacities, respectively.

However, this acceleration of the rotation ages suggested for clonal materials of the species may not be the same for heartwood production, since higher growth rates favor higher sapwood production due to the need for a constant flow of mineral nutrients to assist in photosynthesis (production of photo-assimilates). In view of this and based on the results, we accept the second hypothesis of this study, which states that there is a positive variation in the share of heartwood in the composition of the total volume as the trees mature. However, the heartwood growth did not follow the rate of the volume of the with-bark production of the trees, represented by the noncrossing of the *CAI* and *MAI* curves of the heartwood production, up to 12 years. In view of this, it is important to continue evaluating heartwood evolution at later ages, since the theoretical stagnation in the bark production growth was different to the heartwood growth and formation.

Heartwood tapering mixed models incorporating age as a random effect hold significant importance, particularly for highly valuable woods like teak. Heartwood plays a crucial role in determining the value of such woods, making it essential for the timber industry to comprehend and predict the heartwood growth pattern. In clonal teak plantations, although the overall growth is accelerated, the growth of heartwood does not

follow this pattern proportionally. Consequently, even if a tree has reached a commercially viable diameter with bark, and the apex of the average increment in the total or commercial volume indicates the technical rotation point, it may be prudent to wait a little longer before harvesting. This is because the heartwood continues to increase in size and, therefore, the value of the wood will keep rising as the heartwood proportion in the tree increases.

The utilization of heartwood tapering mixed models in this research, incorporating age as a random variable, facilitated a more precise characterization of future heartwood development concerning the tree diameter. These models established a connection between the tree age and the proportion of heartwood to wood, furnishing valuable information for decision making in clonal teak plantation management and offering more accurate guidelines for harvest planning. Forest owners and the timber industry can employ these models to gain a clearer understanding of when the wood will attain its maximum potential for value in the industrial process. This discovery helps prevent premature harvests and ensures the maximum economic return from plantations with clones of the species in the region, optimizing the estimated stem diameter.

## 5. Conclusions

Based on the models adjusted in this investigation, the Demaerschalk model showed the greatest accuracy in estimating diameters with bark, without bark, and heartwood, as dummy variables, with errors of 1.532 cm and 8.634%. Including age as a random effect in the mixed models positively impacted the accuracy and significance of the diameter estimates along the teak stem, improving the diameter estimation accuracy by 7.2%. The mixed model estimates reveal a gradual and consistent increase in the proportion of heartwood (14 to 34%) as the trees matured, while the bark (23 to 20%) and sapwood (63 to 45%) dimensions showed an inverse pattern. The rate of heartwood growth was found to be distinct from the total or commercial volume, enhancing the importance of considering heartwood increment when making decisions regarding plantation interventions. The employed tapering modeling proved to be an efficient and realistic tool for describing heartwood dimensions and their evolution in clonal teak stands. This modeling provides valuable information for the appropriate planning of harvests, offering essential data to optimize the value of wood through the optimization of product dimension (i.e., length and diameter).

**Author Contributions:** Conceptualization, M.L.d.S., E.P.M., L.J.B., H.J.d.S. and C.R.C.d.S.; methodology, M.L.d.S., E.P.M. and L.J.B.; software, M.L.d.S.; validation, M.L.d.S., H.J.d.S. and C.R.C.d.S.; formal analysis, M.L.d.S., H.J.d.S. and C.R.C.d.S.; investigation, M.L.d.S.; resources, M.L.d.S., E.P.M. and L.J.B.; data curation, M.L.d.S.; writing—original draft preparation, M.L.d.S.; writing—review and editing, M.L.d.S., E.P.M. and L.J.B.; visualization, M.L.d.S., E.P.M., H.J.d.S., C.R.C.d.S., E.A.T.M. and L.J.B.; supervision, E.P.M., E.A.T.M. and L.J.B.; project administration, M.L.d.S. All authors have read and agreed to the published version of the manuscript.

**Funding:** This research received no external funding.

**Data Availability Statement:** The data are not publicly available because of the policy of the company (Tietê Agrícola Ltd.a) that owns the teak plantations.

**Acknowledgments:** We thank the Coordination for the Improvement of Higher Education Personnel (CAPES) and to the National Council for Scientific and Technological Development (CNPq) for providing scholarships. We are grateful to the company Tietê Agrícola Ltd.a for making the study area available and for providing logistical support, and we also thank the University of Brasilia (UnB, Brazil) for the research support.

**Conflicts of Interest:** The authors declare no conflict of interest. The funders had no role in the design of the study; in the collection, analyses, or interpretation of data; in the writing of the manuscript; or in the decision to publish the results.

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
