# Peer review of "The Effect of Age on the Evolution of the Stem Profile and Heartwood Proportion of Teak Clonal Trees in the Brazilian Amazon"

_forests, doi:10.3390/f14101962_

Round 1
Reviewer 1 Report
1. The authors attempted to construct stem profile models to predict diameter outside and inside bark, and heartwood diameter for teak clonal trees. In fact, these models do not show compatibility in prediction, so it is suggested to build a compatible model system to estimate bark, bark-free and heartwood diameters.
2. The authors want to add age to the existing model, and suggests that the author adopt the method of mixed effect model to introduce age variables through random effects. Or adopt a re-parameterization method.
3. In Table 2, why these four taper models were chosen? All four models are simple taper models. As we know many published papers show that Kozak (2004) and Max & Burkhart (1976) are the best models to many species?
4. In Table 2, there is an error in Model 4 where two right brackets are missing.
![]()
5. In line 206, the current annual increment is abbreviated as CAI, but why is equation (2) abbreviated as ICA and why is it also shown as ICA in line 361?
6. In Table 3, the covariate was added incorrectly. t should be in parentheses.

7. In Figures 3a, 4a, and 5a, why is the y-axis in %? How are the percentage estimation errors calculated? Please explain the percentage estimation errors in detail.
8. Some abbreviations should be consistent.

Author Response
Attached is the reply letter to the reviewer.

Reviewer 2 Report
The paper The effect of age on the evolution of the stem profile and on 2 heartwood proportion of teak clonal trees in Brazilian Amazon is fresh and interesting investigation, which completely corresponds to the scope of MDPI journal Forests. The paper is written at good technical level. Obtained results are presented clearly, but the following minor modifications can be done.
1. Chapter "Conclusions" should be expanded and supplied by the numerical results, obtained in course of current study and enables to base conclusions are formulated.
2. Some old sources (No 33, 34, 36, 38, 39, 41 and 44) can be deleted from the list of references.
Author Response

(The authors gave the same response as above.)

Round 2
Reviewer 1 Report
I agree to publish this paper.